# Effectiveness of analgesic ear drops as add-on treatment to oral analgesics in children with acute otitis media: study protocol of the OPTIMA pragmatic randomised controlled trial

Joline L H de Sévaux [1], Roger A M J Damoiseaux,[1] Saskia Hullegie [1],
Elisabeth A M Sanders,[2,3] G Ardine de Wit,[1,4] Nicolaas P A Zuithoff,[1]
Lucy Yardley,[5,6] Sibyl Anthierens,[7] Paul Little [8], Alastair D Hay,[9]
Anne G M Schilder,[1,10,11] Roderick P Venekamp [1]

For numbered affiliations see end of article.

**Correspondence to**
Joline L H de Sévaux;
j.l.h.desevaux-2@umcutrecht.nl

## ABSTRACT

**Introduction** Ear pain is the most prominent symptom of childhood acute otitis media (AOM). To control the pain and reduce reliance on antibiotics, evidence of effectiveness for alternative interventions is urgently needed. This trial aims to investigate whether analgesic ear drops added to usual care provide superior ear pain relief over usual care alone in children presenting to primary care with AOM.

**Methods and analysis** This is a pragmatic, two-arm, individually randomised, open, superiority trial with cost-effectiveness analysis and nested mixed-methods process evaluation in general practices in the Netherlands. We aim to recruit 300 children aged 1–6 years with a general practitioner (GP) diagnosis of AOM and ear pain. Children will be randomly allocated (ratio 1:1) to either (1) lidocaine hydrochloride 5 mg/g ear drops (Otalgan) one to two drops up to six times daily for a maximum of 7 days in addition to usual care (oral analgesics, with/without antibiotics); or (2) usual care. Parents will complete a symptom diary for 4 weeks as well as generic and disease-specific quality of life questionnaires at baseline and 4 weeks. The primary outcome is the parent-reported ear pain score (0–10) over the first 3 days. Secondary outcomes include proportion of children consuming antibiotics, oral analgesic use and overall symptom burden in the first 7 days; number of days with ear pain, number of GP reconsultations and subsequent antibiotic prescribing, adverse events, complications of AOM and cost-effectiveness during 4-week follow-up; generic and disease-specific quality of life at 4 weeks; parents' and GPs' views and experiences with treatment acceptability, usability and satisfaction.

**Ethics and dissemination** The Medical Research Ethics Committee Utrecht, the Netherlands, has approved the protocol (21-447/G-D). All parents/guardians of participants will provide written informed consent. Study results will be submitted for publication in peer-reviewed medical journals and presented at relevant (inter)national scientific meetings.

**Trial registration** The Netherlands Trial Register: NL9500; date of registration: 28 May 2021. At the time of publication of the study protocol paper, we were unable

## STRENGTHS AND LIMITATIONS OF THIS STUDY

⇒ The pragmatic, open design enhances the applicability of the trial findings in daily practice.
⇒ The pragmatic, open design is most suited to address the key secondary outcomes of the impact of the analgesic ear drops on the proportion of children consuming antibiotics and its cost-effectiveness in everyday practice.
⇒ The combination of the open design and the subjective, parent-reported outcomes does not control for the placebo effects, but will more closely reflect the benefits that would accrue in everyday practice with regard to analgesic effects.
⇒ Contamination is possible given the over-the-counter availability of analgesic ear drops.

to make any amendments to the trial registration record in the Netherlands Trial Register. The addition of a data sharing plan was required to adhere to the International Committee of Medical Journal Editors guidelines. The trial was therefore reregistered in ClinicalTrials.gov (NCT05651633; date of registration: 15 December 2022). This second registration is for modification purposes only and the Netherlands Trial Register record (NL9500) should be regarded as the primary trial registration.

## INTRODUCTION

Although the current COVID-19 pandemic has a profound influence on the incidence of acute otitis media (AOM) in children,[1–3] it will remain an important cause of primary care consultations.[4 5] Ear pain is the predominant symptom and central to children's and parents' experience of the condition.[6 7] High-quality studies have shown that most symptoms of AOM settle within a few days without antibiotics in otherwise healthy children living in high-income countries,[8] but it may

take 8 days for ear pain to fully resolve in 90% of the children.[9] Globally, most AOM clinical practice guidelines therefore recommend oral analgesics in all, and antibiotics in selected, children.[10] However, antibiotics are frequently prescribed in children with AOM,[4 5] leading to side effects in at least 1 out of 10 children and emergence of antimicrobial resistance.[11 12] Evidence of effectiveness of management options to relieve the most distressing symptom of AOM and reduce reliance on antibiotics is therefore urgently needed.[13]

Recently, we found that a general practitioner (GP)-targeted educational intervention to improve pain management in children with AOM resulted in an increase in oral analgesic use, in particular ibuprofen, but failed to reduce parent-reported ear pain and antibiotic use.[14] Therefore, alternative interventions need further exploration. Analgesic ear drops, widely available, cheap and safe, may be well suited to fill this gap. When applied into the external auditory canal, these agents could relieve pain and distress by blocking nerve conduction of the tympanic membrane.[15] So far, evidence on their use in children with AOM is limited.[15 16] Our trial aims to produce this important evidence.

## OBJECTIVE
The primary aim of this pragmatic randomised controlled trial is to investigate whether analgesic ear drops added to usual care (oral analgesics with/without antibiotics) provide superior ear pain relief over usual care alone in children presenting to primary care with AOM and ear pain.

## METHODS AND ANALYSIS
### Study design and setting
A pragmatic, two-arm, individually randomised, open, superiority trial with cost-effectiveness analysis in general practices in the Netherlands with a follow-up of 4 weeks will be performed. Potential trial participants will be recruited from primary care (approximately from 150 GPs) in the region of Utrecht, the Netherlands.

A mixed-methods process evaluation will be nested within the trial (see online supplemental material 1).[17–22] This process evaluation will capture data to understand how participating in the trial has an impact on the consultation and behaviour and how AOM interventions (including analgesic ear drops, antibiotics and oral analgesics) are viewed by parents and GPs. The process evaluation consists of a survey with brief statements measuring attitudes of all parents based on the extended Common Sense Model[17] and a three-item Consultation Satisfaction Questionnaire,[18] and a qualitative part (semistructured interviews capturing parents and GPs' views and experiences, respectively).

### Participants
Children aged 1–6 years presenting to primary care with AOM[23] and parent-reported ear pain in 24 hours prior

---

**Box 1  Full list of inclusion and exclusion criteria**

**Inclusion criteria:**
⇒ Age 1–6 years.
⇒ Parent-reported ear pain in 24 hours prior to enrolment.
⇒ General practitioner diagnosis of (unilateral or bilateral) acute otitis media.

**Exclusion criteria:**
Children:
⇒ With (suspected) tympanic membrane perforation or ventilation tubes.
⇒ With ear wax obscuring visualisation of the tympanic membrane.
⇒ Who are systemically very unwell or require hospital admission (eg, child has signs and symptoms of serious illness and/or complications such as mastoiditis/meningitis).
⇒ Who are at high risk of serious complications including children with known immunodeficiency other than partial IgA or $IgG_2$ deficiencies, craniofacial malformation including cleft palate, Down syndrome and previous ear surgery (with the exception of ventilation tubes in the past).

---

to enrolment will be eligible for trial participation. Children with (suspected) tympanic membrane perforation or ventilation tube (as they carry a risk of inner ear damage causing hearing loss or tinnitus after application of lidocaine drops), ear wax obscuring visualisation of the tympanic membrane or those who are very unwell or require hospitalisation will be excluded. Detailed inclusion and exclusion criteria are listed in box 1.

### Study procedures and data collection
#### Recruitment and inclusion of participants
Figure 1 depicts a flow diagram of the study procedures. During regular primary care consultations, participating GPs will inform parents of potentially eligible children about the trial. Next, GPs will ask parents who are interested in participation for consent to provide their contact details to the study team at the University Medical Center (UMC) Utrecht. Members of the team will contact the parents by phone the same day to provide detailed information about the trial. If parents provisionally agree to participate and if the child fulfils the trial eligibility criteria, the study physician will visit the child at home. At this home visit, written informed consent for participation in the trial and the semistructured interview will be obtained (online supplemental material 2), baseline demographic and disease-specific data will be collected and a physical examination including otoscopy will be performed. Parents are also asked to complete generic and disease-specific quality of life questionnaires on behalf of their child. A full description of the nested process evaluation is given in online supplemental material 1.

#### Study group assignment
At the conclusion of the home visit, a trial randomisation website will be accessed for concealed study group assignment. An independent data manager generates a

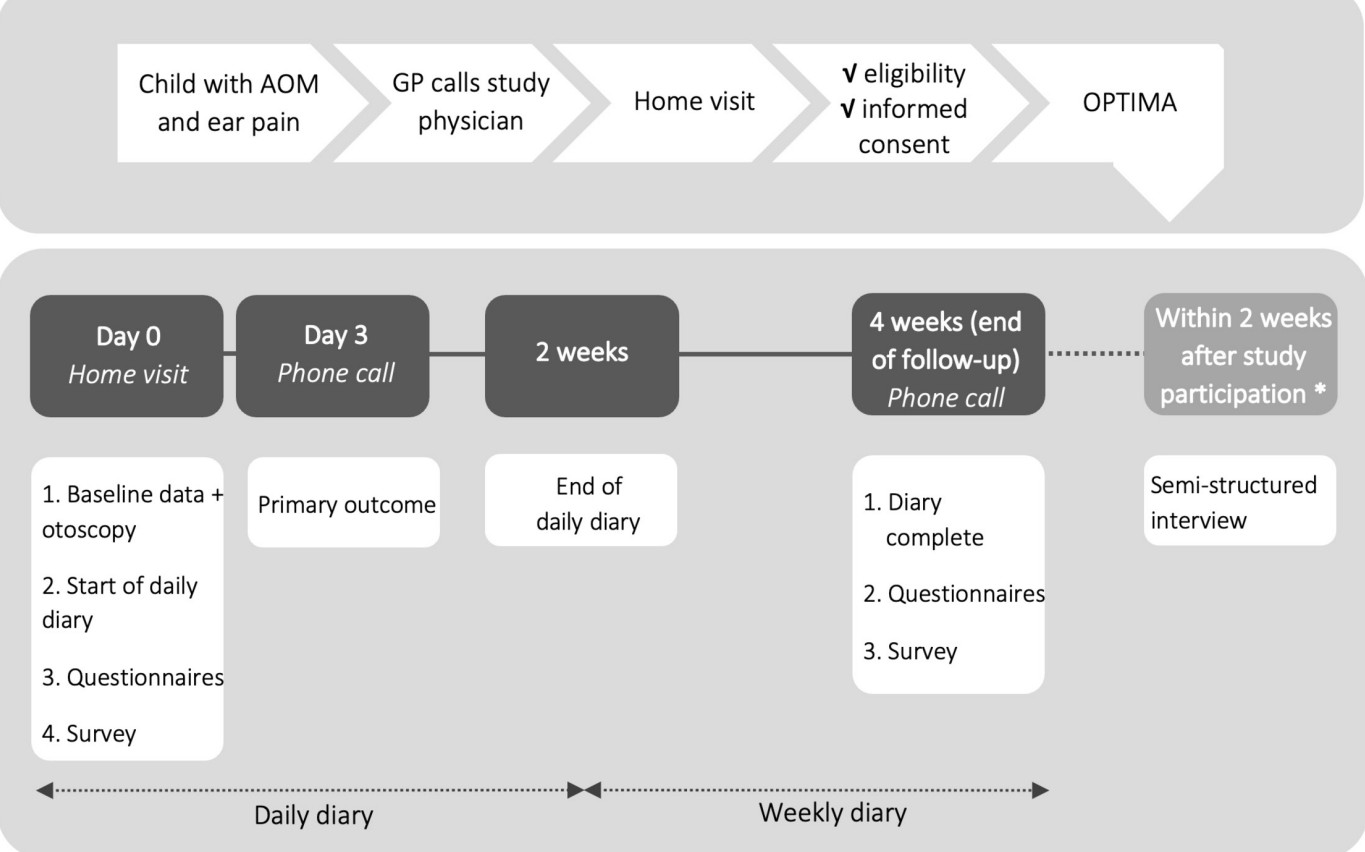

**Figure 1** Flow diagram of study procedures. *Parents who gave consent for the semistructured interview will be recruited from both the intervention and control arm and will be purposively sampled after follow-up. AOM, acute otitis media; GP, general practitioner; OPTIMA, trial acronym.

computer-generated randomisation sequence stratified for age (<2 years vs 2 years and above), AOM laterality (unilateral vs bilateral) and AOM baseline antibiotic prescribing (yes vs no). Children will be randomly allocated to either (1) lidocaine hydrochloride 5 mg/g (Otalgan) ear drops one to two drops up to six times daily for a maximum of 7 days[24 25] with usual care (oral analgesics, with/without antibiotics); or (2) usual care. Analgesic ear drops will be provided by the study team. These drops should not be used in children with a tympanic membrane perforation or ventilation tube as they carry a risk of inner ear damage causing hearing loss or tinnitus. These children will therefore be excluded, as well as those with ear wax obscuring visualisation of the tympanic membrane. Parents of children allocated to the intervention group are expressly instructed to stop applying the drops if their child develops ear discharge.

The study team will notify the GP about the result of the randomisation. The local pharmacist will be notified in case the child is allocated to the analgesic ear drops, in order to check any known sensitivities/allergies and any interactions (with current or future drugs). Any treatment decisions other than the use of analgesic ear drops, that is, antibiotics and oral analgesics, will be left to the GP's discretion in both groups. To those allocated to the intervention group, the study physician will not provide

any treatment advice other than instructions about the use of analgesic ear drops. To those allocated to the control group, the study physician will not provide any treatment advice.

During follow-up, parents and GPs will be encouraged to manage AOM symptoms according to current clinical practice guidance on AOM in children issued by the Dutch College of General Practitioners. Lidocaine ear drops are currently not recommended in this guideline due to the lack of evidence.[23]

### Follow-up data collection

Participants will be followed for 4 weeks. A telephone call will be scheduled at day 3, which will allow us to capture critical data on our primary outcome. Parents will keep a diary of AOM-related symptoms including ear pain scores and fever recordings, treatment adherence, prescribed and over-the-counter (OTC) medication use, AOM-related healthcare resource use, child care and travel costs, adverse events and complications of AOM for 4 weeks. At the end of follow-up, parents will complete a questionnaire on parental productivity losses and generic and disease-specific quality of life questionnaires. The diary and questionnaires can either be completed online or on paper forms (see the Data management section). The study team will contact the parents by phone to

check if all data are collected. During this phone call, an appointment for the semistructured interview can be made with parents who are still willing to participate in the nested qualitative investigation.

### Validated questionnaires used in the trial

► Child's ear pain intensity will be reported by parents in the symptom diary using a 0–10 Numerical Rating Scale, which has been validated for measuring pain in children with AOM.[26]
► Child's symptoms of crying/distress, disturbed sleep, interference with normal activity, appetite, fever and hearing problems will be reported by parents in the symptom diary using a 0–6 Likert scale with higher scores indicating greater severity. This scale has been proven valid and was successfully used in our previous trials.[16 26]
► Generic quality of life of the child at baseline and 4 weeks will be assessed using the 47-item short-form of the Infant Toddler Quality of Life Questionnaire, which is developed for use in infants and toddlers aged 2 months–6 years and has shown to be a feasible instrument with adequate psychometric properties to discriminate between children with and without doctor-diagnosed respiratory disease.[27]
► Disease-specific quality of life of the child at baseline and 4 weeks will be assessed using the Otitis Media-6 Questionnaire.[28]
► Productivity losses (eg, parental absenteeism from work) because of AOM of their child will be assessed using the IMTA Productivity Cost Questionnaire (iPCQ) at 4 weeks.[29]

### Outcomes
#### Primary outcome
The parent-reported ear pain score over the first 3 days. To this end, parents will record their child's ear pain scores during the first 3 consecutive days using a 0–10 validated Numerical Rating Scale.

#### Secondary outcomes
► Proportion of children consuming antibiotics in the first 7 days.
► Oral analgesic use in the first 7 days.
► Overall symptom burden (crying/distress, disturbed sleep, interference with normal activity, appetite, fever and hearing problems) using a 0–6 Likert scale in the first 7 days.
► Number of days with ear pain during follow-up (4 weeks).
► Number of GP reconsultations with/without subsequent antibiotic prescribing during follow-up.
► Adverse events during follow-up.
► Complications of AOM during follow-up.
► Costs and cost-effectiveness during follow-up.
► Generic quality of life of the child at 4 weeks.
► Disease-specific quality of life of the child at 4 weeks.

► Parents' and GPs' views of treatment acceptability, usability and satisfaction.

### Sample size calculation
The main aim is to demonstrate that analgesic ear drops as added to usual care provide superior parent-reported ear pain relief over the first 3 days compared with usual care. In our recent trial, the mean parent-reported ear pain score (0–10 scale) over the first 3 days was 4.36 (SD 2.12) among those allocated to usual care.[14] Patient and public involvement input from the UK suggested that a 1-point reduction (on a 0–10 scale) would be a clinically important difference.[16] However, our parent panel felt that even a somewhat smaller difference would be clinically meaningful given the low-cost, low-risk nature of the intervention. To detect a pain reduction of 20% (reflecting a 0.87-point reduction in pain) compared with usual care with 90% power (beta 0.1) and at a 5% significance level (alpha 0.05), 126 children per group will be needed. To allow for 20% attrition, we aim to include 300 participants.

The estimated sample size will also provide substantial power to address the key secondary outcome of the proportion of children consuming antibiotics. Anticipating 47% of children consuming antibiotics in the usual care group as with our previous trial,[14] 252 children will provide 89% power (beta 0.11; alpha 0.05) to detect a 20% difference in the proportion of children consuming antibiotics.

### Data management
During study conduct, we will comply with the European Union General Data Protection Regulation and the Dutch Act on Implementation of the General Data Protection Regulation. The data management department of the Julius Center, UMC Utrecht will be responsible for handling and storage of the trial data using a secured and coded online database (Research Online 2.0 (RO 2.0)). Data will be either entered directly into RO 2.0 or collected on paper forms first. For data entered directly into RO 2.0, processes to promote data quality are added. Coded trial data will be securely stored for 25 years in the main study folder on a central drive of the Julius Center.

### Patient and public involvement
A panel of eight parents was established for this trial. This panel helped to shape the grant proposal and final study protocol during online focus group meetings, with particular advice given on patient-relevant outcomes including the minimally important clinical differences, recruitment strategy and the design of data collection. The parent panel will be actively involved throughout all critical stages of the trial through regular parent panel meetings.

### Statistical analysis
All analyses will be performed according to the intention-to-treat principle and blinded with respect to study group assignment. The results will be reported in accordance

with the Consolidated Standards of Reporting Trials guidelines.[30]

## Clinical effectiveness

Descriptive statistics will be used to assess any baseline differences between treatment groups. The primary outcome will be analysed with a linear regression model with a residual covariance (ie, Generalized Estimating Equations (GEE) type) matrix to correct for repeated measurements. In main (adjusted) analysis, treatment group, time, a priori specified prognostic factors (baseline ear pain score, duration of ear pain prior to enrolment) and stratification factors (age (<2 years vs 2 years and above), AOM laterality (unilateral vs bilateral) and baseline antibiotic prescribing (yes vs no)) will be included in the model. The validity of the model, that is, normality and homoscedasticity, will be evaluated by assessing residuals and the comparison between treatment groups will be expressed as differences in means with 95% CIs. If more than 5% of baseline and outcome data are missing, a sensitivity analysis will be performed using multiple imputation techniques.[31 32]

Secondary outcomes will be analysed using negative-binomial regression analyses for count variables, log binomial regression analyses for dichotomous variables and linear regression analyses for continuous variables, where applicable, adjusted for repeated measurements. For these analyses, the comparison between treatment groups will be expressed as rate ratios, risk ratios and mean differences, respectively, all with 95% CIs.

We will also conduct prespecified, exploratory subgroup analyses to investigate whether age (<2 years vs 2 years and above) or AOM laterality (unilateral vs bilateral) modifies the effectiveness of analgesic ear drops. In further sensitivity analysis, we will also explore the added impact of antibiotic consumption on the primary outcome of interest.

## Cost-effectiveness analysis

Alongside the trial, a cost-effectiveness analysis from a societal perspective, that is, including both medical and non-medical costs, will be performed. The societal perspective is particularly relevant in this case, since non-medical costs are particularly relevant in AOM.[33] Those costs mainly relate to parental productivity losses and costs of additional childcare.

## Cost analysis

All costs will be estimated at the patient level. Costs of medication use will be retrieved from the Dutch formulary and a pharmacist's fee will be added for every prescription.[24 34] Costs of OTC and complementary medicines will be calculated per day, based on current average retail prices. Costs of consulting a GP or a medical specialist, and costs of hospitalisation will be based on current Dutch guidelines for pharmaco-economic evaluation in healthcare research. Costs of diagnostic tests will also be derived from these guidelines.[34 35] Costs of surgical procedures

will be retrieved from a previous Dutch costing study that calculated costs for the different components of surgical procedures.[36]

Resource use such as doctor's visits, prescribed medication (including antibiotics), specialist referrals, hospital admissions, surgical interventions as well as out-of-pocket expenses such as OTC medication, child care and travel costs will be collected from study diaries filled out by parents. Cost prices will be estimated according to guidelines for pharmaco-economic evaluation in healthcare research.[34 35] Costs associated with productivity losses will be retrieved from the iPCQ Questionnaire and estimated using the friction cost method.[29 37] The hourly cost estimate for childcare will be derived from the Dutch National Institute for Family Finance Information.[38] Travel expenses will be calculated for healthcare visits following the Dutch guideline for economic evaluation.[35] Mean costs per patient will be compared across the randomisation groups. As the time horizon is only 4 weeks, discounting does not apply.

## Cost-effectiveness results

Costs from a societal perspective will be compared with the parent-reported ear pain scores over the first 3 days. We will calculate incremental cost-effectiveness ratios by dividing the estimated differences in costs between groups by the differences in effects observed, that is, the additional cost per additional 1-point reduction in mean ear pain score over the first 3 days for the analgesic ear drops group versus the usual care group. Uncertainty will be addressed by means of two-stage non-parametric bootstrap sampling. The effect of any difference in baseline characteristics on cost-effectiveness results will be investigated using net benefit regression methods. In case of missing data, we will use multiple imputation techniques. The final results will be presented using incremental cost-effectiveness planes and cost-effectiveness acceptability curves.

## Monitoring and safety

The trial will be monitored regularly by an independent research data monitor of the UMC Utrecht to ensure the quality of the trial execution. Since we do not expect any safety issues and the overall risk of this trial was judged negligible, we have not established a data safety monitoring board and refrain from conducting a safety interim analysis. However, in accordance to section 10, subsection 4, of the Medical Research Involving Human Subjects Act (WMO), the sponsor will suspend the study if there is sufficient ground that continuation of the study will jeopardise subject health or safety.

## ETHICS AND DISSEMINATION

The study will be conducted according to the principles of the Declaration of Helsinki (10th version, October 2013), the International Conference on Harmonisation (ICH) Good Clinical Practice E6(R2) guideline, the Dutch Law

on medicinal products ('geneesmiddelenwet') and in accordance with the WMO. The Medical Research Ethics Committee Utrecht, the Netherlands, has approved the protocol (21-447/G-D).

Research findings will be disseminated through publications in peer-reviewed medical journals targeted towards the wider audience of health professionals involved in the care of children and presented at relevant national and international scientific meetings. We will work with our parent panel to help interpret the findings of the trial and harness their resources for dissemination to the lay public.

On completion of the trial, data will be stored for a minimum of 25 years at a central data drive at the Julius Center. We will publish the metadata in the DataverseNL repository and will provide a persistent identifier. The data will be available to researchers who provide a methodologically sound proposal to achieve the aims in the approved proposal. Proposals to gain access to the data should be directed to the principal investigator (RV). Data requestors will need to sign a data sharing agreement.

## CURRENT STUDY STATUS

Study enrolment is ongoing with the first participant being included on 6 October 2021.

## DISCUSSION

Up to now, evidence on the use of analgesic ear drops in children with AOM is limited. A 2011 systematic review included five studies of which two (117 children) compared analgesic ear drops with placebo.[15] The authors concluded that these trials provide limited evidence that a single dose of analgesic ear drops is effective in reducing ear pain 30 min after administration in older children (age 3–19 years) with AOM.[15] Since then, a UK-based trial compared at-home, repeated dosing, of analgesic ear drops with placebo ear drops and usual care. This trial was terminated prematurely due to operational issues.[16] Initial data of 84 children recruited in primary care indicate that ear pain levels were lower in both ear drops groups than in the usual care group. However, these estimated treatment effects are imprecise because of the low sample size target and therefore require confirmation in future studies.

Our trial aims to do so by assessing the clinical and cost-effectiveness of analgesic ear drops in children with AOM, as well as views and experiences of parents and GPs through the nested process evaluation. The pragmatic, open design enhances the applicability of the trial findings to daily practice and is most suited to address the key secondary outcome of the impact of the analgesic ear drops on the proportion of children consuming antibiotics in everyday practice.[39] This would be much more difficult to determine realistically in a placebo-blinded study where children in both groups would receive ear drops (either placebo or analgesic drops); such 'treatment' may in itself lead to altered parental (healthcare seeking) behaviour. Additionally, the open design, in which the control group receives usual care, is most suited for 'real-world' cost-effectiveness analysis. However, the lack of blinding in combination with the subjective, parent-reported outcomes might also lead to bias in the assessment of analgesic effects. To minimise this potential bias, the study physician will emphasise during the baseline visit that it is not known whether the drops have any effect in children with AOM. Furthermore, a recent meta-epidemiological study found no evidence for a difference in estimated treatment effect between blinded and non-blinded trials.[40] Although it is still unclear what caused the results of this study, it suggests that blinding is less important than anticipated. Our design does not control for the placebo effects, but will more closely reflect the benefits that would accrue in everyday practice with regard to analgesic effects.

Although not recommended in the Dutch AOM clinical practice guideline,[23] the OTC availability of lidocaine ear drops may inherently incur the risk of contamination, that is, parents in the control group buying lidocaine ear drops themselves. We consider this unlikely since only 5% (10 of 221) of participants did use lidocaine ear drops in our previous trial of a GP-targeted educational intervention to improve pain management in children with AOM.[14] We will, however, capture data on the use of lidocaine ear drops in all trial participants to monitor potential contamination.

**Author affiliations**
[1]Julius Center for Health Sciences and Primary Care, University Medical Center Utrecht, Utrecht, The Netherlands
[2]Department of Paediatric Immunology and Infectious Diseases, Wilhelmina Children's Hospital University Medical Center, Utrecht, The Netherlands
[3]Centre for Infectious Disease Control, National Institute for Public Health and the Environment (RIMV), Bilthoven, The Netherlands
[4]Centre for Nutrition, Prevention and Healthcare, National Institute for Public Health and the Environment (RIVM), Bilthoven, The Netherlands
[5]School of Psychology, Faculty of Environmental and Life Sciences, University of Southampton, Southampton, UK
[6]School of Psychological Science, Faculty of Life Sciences, University of Bristol, Bristol, UK
[7]Department of Family Medicine and Population Health, Faculty of Medicine and Health Sciences, University of Antwerp, Antwerp, Belgium
[8]Primary Care Research Centre, Primary Care Population Sciences and Medical Education, Faculty of Medicine, University of Southampton, Aldermoor Health Centre, Southampton, UK
[9]Centre for Academic Primary Care, Bristol Medical School: Population Health Sciences, University of Bristol, Bristol, UK
[10]Biomedical Research Centre, NIHR University College London Hospitals, London, UK
[11]evidENT, Ear Institute, University College London, London, UK

**Acknowledgements** We gratefully thank our parent panel for their valuable input and all children and parents who participated in our trial thus far.

**Contributors** RV, AS and RAMJD designed the trial. JLHdS, SH and RV drafted the first version of the protocol paper. All other authors (RAMJD, AS, ES, AAW, NPAZ, LY, SA, PL and ADH) provided feedback on this version. All authors (JLHdS, SH, RV, RAMJD, AS, ES, AAW, NPAZ, LY, SA, PL and ADH) approved the final version of the manuscript.

**Funding** The trial is supported by a grant from the Netherlands Organisation for Health Research and Development (ZonMw–grant number 10060011910003). LY is an NIHR Senior Investigator and her research programme is partly supported by NIHR Applied Research Collaboration (ARC)-West, NIHR Health Protection Research Unit (HPRU) for Behavioural Science and Evaluation, and the NIHR Southampton Biomedical Research Centre (BRC).

**Disclaimer** The funding agency had no role in the design, and will not have any role during execution of the trial, data analyses, interpretation of the data or decision to submit results.

**Competing interests** None declared.

**Patient and public involvement** Patients and/or the public were involved in the design, or conduct, or reporting, or dissemination plans of this research. Refer to the Methods section for further details.

**Patient consent for publication** Not required.

**Provenance and peer review** Not commissioned; externally peer reviewed.

**ORCID iDs**
Joline L H de Sévaux http://orcid.org/0000-0002-0472-3455
Saskia Hullegie http://orcid.org/0000-0003-3413-1423
Paul Little http://orcid.org/0000-0003-3664-1873
Roderick P Venekamp http://orcid.org/0000-0002-1446-9614

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
