## [Reviewer comments · BMJ Open]

ARTICLE DETAILS

TITLE (PROVISIONAL)	Effectiveness of analgesic ear drops as add-on treatment to oral analgesics in children with acute otitis media: study protocol of the OPTIMA pragmatic randomised controlled trial
AUTHORS	de Sévaux, Joline; Damoiseaux, Roger; Hullegie, Saskia; Sanders, Elisabeth; Wit, Ardine; Zuithoff, Nicolaas; Yardley, Lucy; Anthierens, Sibyl; Little, Paul; Hay, Alastair; Schilder, Anne; Venekamp, Roderick

VERSION 1 – REVIEW

REVIEWER	Juan Dewez London School of Hygiene and Tropical Medicine, Clinical Research Department
REVIEW RETURNED	22-Mar-2022

GENERAL COMMENTS	Dear Authors, Your protocol is very clear and well-written. I have no comments nor queries.
---

REVIEWER	Amanda Leach Menzies School of Health Research, Child Health Division
REVIEW RETURNED	31-Mar-2022

GENERAL COMMENTS	Effectiveness of analgesic ear drops as add-on treatment to oral analgesics in children with acute otitis media: study protocol of the OPTIMA pragmatic randomised controlled trial. ID: 2022-062071 This is a very challenging research question to address, with few trials and very low participant numbers. I congratulate the research team for whom I have a very high regard in the field. The first question of reviewers is to ensure study dates are included. I did find this was 6 October 2021 (page 12). This information could be included earlier in the manuscript. The major issue that I have with this trial is the rationale for no topical placebo. Page 13. In the discussion the rationale for and against use of a placebo in trials of analgesic drops is presented. Placebo-controlled trials have been conducted of single dose and repeat dose, leading to a limited body of evidence. The authors have chosen an open design (without placebo) which is believed to be more applicable to daily practice, since placebo 'treatment' may lead to altered parental behaviour. Yet there is also an admission that lack of blinding in combination with subjective outcomes increases bias. To counter this the study physician will inform parents of the uncertainty about effectiveness of drops in children with AOM. Finally they cite a
---

	recent meta-epidemiological study that found no evidence for a difference in estimated treatment effect between blinded and non-blinded trials. However, this is no evidence for a difference, not evidence of no difference, and the final conclusion of Moustgaard et al. is that “blinding should remain the safeguard in trials.” As the trial has already commenced, inclusion of a third arm with placebo could be difficult, however it is strongly recommended. Alternatively, is there any intention to include an interim analysis? Page 2 line 23: add ratio 1:1 Page 4 Line 25: I note Trial NL4781 (NTR4920) 2015- 2019 (finish date) of Pain Intensity Monitoring in Paediatric Otitis Media NTR (trialregister.nl) Will GPs in this trial have been in the PIMPOM study of pain management? Any stratification if a proportion of GPs were in prior study? Page 6: Parents are asked to complete a disease-specific QoL questionnaire on day0, how will this influence parent-reported pain in an open label trial? Supplement: There are baseline and 4-week timepoints for the process evaluation (supplement) however the trial should attempt to minimise contamination and consider removing the baseline questionnaire. Supplement: Semi-structured interviews are planned after study participation, within 2 weeks. GPs will be interviewed “towards the end of the inclusion period” – this needs to be more specific as they will be continuously referring participants to the trial. More details of the baseline, semi-structured interviews, and 4-week follow-up questionnaires, satisfaction questionnaires are needed to make clear the risk of contamination or other influence on parent reported outcomes. When will parents complete the satisfaction questionnaires? Could all interview/questionnaire timepoints and more detail be included in the flow diagram? Are questionnaires completed on paper, on-line, or by phone interview conducted by whom? Page 6: Box 1. Age 1 to 6 years Generally, children < 2 years of age are at high risk of poor treatment outcomes, will parents consent to enrolling this age group? I note stratification by age <2/>2, how will you ensure adequate power for stratified groups? As the age-range includes school-age children is loss of days at school an important outcome? This should be a fairly easy objective measure of interest to parents. Page 7 line 28 “To those allocated the intervention group, the study physician will not provide any treatment advice other than .. drops.” As the same study person is obtaining informed consent and conducting the baseline questionnaires with parents at this time, this will be difficult. Why is this restriction (to not provide advice) only placed on the intervention group? Topical analgesics are not recommended unless administered by health professional (in case of TM perforation and risk of ototoxicity). In your study, children with (suspected) TMP or TTs are to be excluded. Will parents (intervention and control groups) be advised to notify study physician if they see ear discharge? Please explain why pharmacist notified of randomisation? Is this to discourage parents allocated to no analgesic ear drops from buying them at their local pharmacy? Page 7 line 32: What are the current Dutch College of GP guidelines regarding topical analgesics? Recommended or not? I see your answer on Page 13, line 53. Generally: There is some inconsistency in the use of tense – “will
--	--

	be” or “were” – also in Supplemental material 1. Page 7/8: Will parents also use the faces scale for ear pain? Page 8 Line 5: Will parents be given a thermometer to record fever? Parents are typically not good at noticing fever, yet a temperature is a good objective measure. Page 8: line 17: change discriminates to discriminate Page 13 Line 60: add ‘in’ as per “...to improve pain management in children with AOM.” Is there any administrative or commercial mechanism for tracking over-the-counter purchasing of topical analgesics?
--	--

VERSION 1 – AUTHOR RESPONSE

Reviewer: 1

Dr. Juan Dewez, London School of Hygiene and Tropical Medicine

Comments to the Author:

Dear Authors,

Your protocol is very clear and well-written.
I have no comments nor queries.

AU: We would like to thank the reviewer for the positive judgement.

Reviewer: 2

Prof. Amanda Leach, Menzies School of Health Research

Comments to the Author:

This is a very challenging research question to address, with few trials and very low participant numbers. I congratulate the research team for whom I have a very high regard in the field.

AU: We would like to thank the reviewer for the kind words.

The first question of reviewers is to ensure study dates are included. I did find this was 6 October 2021 (page 12). This information could be included earlier in the manuscript.

AU: As noticed by the reviewer, we have included study dates in the manuscript. We have adhered to the BMJ Open house style when drafting the protocol paper, but are happy for the copy editor to make any changes if necessary.

The major issue that I have with this trial is the rationale for no topical placebo.

Page 13. In the discussion the rationale for and against use of a placebo in trials of analgesic drops is presented. Placebo-controlled trials have been conducted of single dose and repeat dose, leading to a limited body of evidence. The authors have chosen an open design (without placebo) which is believed to be more applicable to daily practice, since placebo ‘treatment’ may lead to altered parental behaviour. Yet there is also an admission that lack of blinding in combination with subjective outcomes increases bias. To counter this the study physician will inform parents of the uncertainty about effectiveness of drops in children with AOM. Finally they cite a recent meta-epidemiological study that found no evidence for a difference in estimated treatment effect between blinded and non-blinded trials. However, this is no evidence for a difference, not evidence of no difference, and the final conclusion of Moustgaard et al. is that “blinding should remain the safeguard in trials.”

As the trial has already commenced, inclusion of a third arm with placebo could be difficult, however it is strongly recommended. Alternatively, is there any intention to include an interim analysis?

AU: As the reviewer rightly points out, our design does not control for the placebo effects. This has been extensively discussed in our multidisciplinary study team at the design phase of our study. As stated in our manuscript, we believe that the open design will more closely reflect the benefits that would accrue in everyday practice with regard to analgesic effects. However, we cannot ignore the possibility that the lack of blinding in combination with subjective outcomes leads to bias. We try to minimise such effect by emphasizing the uncertainty about the effects of topical analgesics to the parents prior to enrolment of eligible children. Since the trial has already commenced, we cannot make any changes to the design of our study.

Regarding the meta-epidemiological study we cited, the reviewer raises a valid point. No evidence for a difference is not the same as evidence of no difference, however, the results of this study suggest that blinding is less important than anticipated. To avoid ambiguity, we have now added this sentence: "Although it is still unclear what caused the results of this study, it suggests that blinding is less important than anticipated."

We refrain from performing a-priori defined interim analysis for futility or safety. We do not expect any safety issues and the overall risk of this trial was judged negligible by the Medical Research Ethics Committee. Furthermore, given the overall sample size of the trial, the sample size at an interim analysis will unlikely be sufficient to provide sufficient power for reaching any a-priori defined futility or superiority margins.

Page 2 line 23: add ratio 1:1

AU: As suggested, we added 'ratio 1:1' to this sentence.

Page 4 Line 25: I note Trial NL4781 (NTR4920) 2015-2019 (finish date) of Pain Intensity Monitoring in Paediatric Otitis Media NTR (trialregister.nl) Will GPs in this trial have been in the PIMPOM study of pain management? Any stratification if a proportion of GPs were in prior study?

AU: No GPs participating in our trial have also participated in the PIMPOM study.

Page 6: Parents are asked to complete a disease-specific QoL questionnaire on day 0, how will this influence parent-reported pain in an open label trial?

AU: The disease-specific QoL questionnaire used in our trial records ear-related problems in the previous period (so before study participation). Therefore, we don't think it will affect the parent-reported ear pain reported during follow-up. Besides, both groups complete the questionnaires and diary (including parent-reported pain) . It is therefore unlikely that this would affect trial outcomes.

Supplement: There are baseline and 4-week timepoints for the process evaluation (supplement) however the trial should attempt to minimise contamination and consider removing the baseline questionnaire.

AU: At baseline, we ask parents to complete a survey about their beliefs about the necessity and harms of antibiotics for AOM, oral analgesics and analgesic eardrops prior to trial participation. This provides key information about any potential difference in beliefs between groups at baseline. Parents are asked to complete the same survey at 4 weeks (end of follow up). This allows us to see if these beliefs changed after participating in the study. Qualitative data collection by means of semi-structured interviews will be conducted after study participation.

Supplement: Semi-structured interviews are planned after study participation, within 2 weeks. GPs will be interviewed "towards the end of the inclusion period" – this needs to be more specific as they will be continuously referring participants to the trial.

More details of the baseline, semi-structured interviews, and 4-week follow-up questionnaires, satisfaction questionnaires are needed to make clear the risk of contamination or other influence on parent reported outcomes. When will parents complete the satisfaction questionnaires? Could all interview/questionnaire timepoints and more detail be included in the flow diagram? Are questionnaires completed on paper, on-line, or by phone interview conducted by whom?

AU: We agree with the reviewer that it will be better to conduct the semi-structured interviews with the GPs when they do not refer any new participants to the trial (so after the inclusion period). This has now been changed accordingly.

As stated in supplemental material 1, the three-item Consultation Satisfaction Questionnaire is part of the survey including brief statements which will be rated by parents on a 5-point scale (strongly disagree/ strongly agree). Parents of all trial participants will complete this survey at baseline and at 4 weeks (end of follow-up).

The diary, questionnaires and surveys can either be completed online or on paper forms. Phone calls serve as reminders/checks for completing the diary and questionnaires.

We updated the manuscript and the flowchart with the information above.

Page 6: Box 1. Age 1 to 6 years

Generally, children < 2 years of age are at high risk of poor treatment outcomes, will parents consent to enrolling this age group? I note stratification by age <2/>2, how will you ensure adequate power for stratified groups?

AU: The purpose of stratification by age is not to have sufficient power for subgroup analyses, but to ensure balance across the age groups between the intervention and control group (since the prognosis may differ between children aged >2/>2 years).

In our previous trial, we recruited substantial number of children aged <2 years.¹ In addition, oral antibiotics can be used during trial participation if deemed necessary and the analgesic eardrops used in this trial are safe for children >1 year of age. Therefore, we are confident that parents will also consent to enrolling children <2 years of age.

1. van Uum RT, Venekamp RP, Zuithoff NP, et al. Improving pain management in childhood acute otitis media in general practice: a cluster randomised controlled trial of a GP-targeted educational intervention. *Br J Gen Pract* 2020;70(699):e684-e95. doi: 10.3399/bjgp20X712589 [published Online First: 2020/08/26]

As the age-range includes school-age children is loss of days at school an important outcome? This should be a fairly easy objective measure of interest to parents.

AU: In this trial, we collect data on loss of days at school or day care and parental productivity losses. This information will be instrumental for our cost analysis.

Page 7 line 28 “To those allocated the intervention group, the study physician will not provide any treatment advice other than .. drops.” As the same study person is obtaining informed consent and conducting the baseline questionnaires with parents at this time, this will be difficult. Why is this restriction (to not provide advice) only placed on the intervention group?

AU: As stated in the manuscript, any treatment decisions other than the use of analgesic ear drops, i.e. antibiotics and oral analgesics, will be left to the GP’s discretion in both groups. This will also apply to the control group, however parents from the control group do not need any instructions about the use of analgesic ear drops. To avoid ambiguity, we have now added the sentence: “To those allocated to the control group, the study physician will not provide any treatment advice.”.

Topical analgesics are not recommended unless administered by health professional (in case of TM perforation and risk of ototoxicity). In your study, children with (suspected) TMP or TTs are to be excluded. Will parents (intervention and control groups) be advised to notify study physician if they see ear discharge?

AU: Yes, parents are asked to notify the study physician should ear discharge occur. Parents in the intervention group are instructed to stop applying the analgesic eardrops in case of ear discharge. This is stated in the information letter/informed consent form (see supplemental material 2, in Dutch). We have now added this information to the manuscript as well: "These drops should not be used in children with a tympanic membrane perforation or ventilation tube as they carry a risk of inner ear damage causing hearing loss or tinnitus. These children will therefore be excluded, as well as those with ear wax obscuring visualisation of the tympanic membrane. Parents of children allocated to the intervention group are expressly instructed to stop applying the drops if their child develops ear discharge."

Please explain why pharmacist notified of randomisation? Is this to discourage parents allocated to no analgesic ear drops from buying them at their local pharmacy?

AU: Notification of the local pharmacist about the trial result will only take place for those allocated to analgesic ear drops. This a standard procedure; to check any known sensitivities/allergies and any interactions (with current or future drugs). As such, we do not aim to discourage parents in the control group from buying them at their local pharmacy with this procedure. To avoid any misunderstanding, we have now rephrased this sentence to: "The study team will notify the GP about the result of the randomisation. The local pharmacist will be notified in case the child is allocated to the analgesic ear drops, in order to check any known sensitivities/allergies and any interactions (with current or future drugs)".

Page 7 line 32: What are the current Dutch College of GP guidelines regarding topical analgesics? Recommended or not? I see your answer on Page 13, line 53.

AU: Lidocaine eardrops are currently not recommended in the AOM guideline issued by the Dutch College of GPs due to the lack of evidence. This information has now also been added to page 7, line 32.

Generally: There is some inconsistency in the use of tense – "will be" or "were" – also in Supplemental material 1.

AU: We checked and revised the manuscript throughout and where appropriate to ensure consistency in the use of tense.

Page 7/8: Will parents also use the faces scale for ear pain?

AU: No, ear pain will be reported by parents using a 0-10 numerical rating scale only. As stated in the paper, this scale has been validated for measuring pain in children with AOM.

Page 8 Line 5: Will parents be given a thermometer to record fever? Parents are typically not good at noticing fever, yet a temperature is a good objective measure.

AU: Since analgesic eardrops are unlikely to influence the body temperature, we chose not to use temperature as objective outcome measure in this trial. Therefore, parents will not be given a

thermometer. The overall symptom burden include the recording of fever but will be measured using a 0-6 Likert scale (which do not contain the exact body temperature recordings)..

Page 8: line 17: change discriminates to discriminate

AU: We changed the manuscript accordingly.

Page 13 Line 60: add 'in' as per "...to improve pain management in children with AOM."

AU: We added 'in' to this sentence as suggested by the reviewer.

Is there any administrative or commercial mechanism for tracking over-the-counter purchasing of topical analgesics?

AU: Such mechanism is not available in the Netherlands. However, we do ask parents to record the use of any over-the-counter medication (including analgesic eardrops) in the daily diary.

VERSION 2 – REVIEW

REVIEWER	Amanda Leach Menzies School of Health Research, Child Health Division
REVIEW RETURNED	10-Oct-2022
GENERAL COMMENTS	Apologies for delay - it took me a while to see the Author response document as it was not a part of the PDF. Thanks you for your considered responses. Well done and all the best. I have left the score for study design as 'no' in the light of no placebo. As you say, it was strongly debated by the research team so diversity of expert opinion is understandable.

VERSION 2 – AUTHOR RESPONSE

Reviewer: 2

Prof. Amanda Leach, Menzies School of Health Research Comments to the Author:

Apologies for delay - it took me a while to see the Author response document as it was not a part of the PDF. Thanks you for your considered responses. Well done and all the best. I have left the score for study design as 'no' in the light of no placebo. As you say, it was strongly debated by the research team so diversity of expert opinion is understandable.

Reviewer: 2

Competing interests of Reviewer: none

AU: We would like to thank the reviewer.